# MicroRNA Biomarkers as Promising Tools for Early Colorectal Cancer Screening—A Comprehensive Review

**DOI:** 10.3390/ijms241311023

**Published:** 2023-07-03

**Authors:** Daniela A. R. Santos, Cristiana Gaiteiro, Marlene Santos, Lúcio Santos, Mário Dinis-Ribeiro, Luís Lima

**Affiliations:** 1Experimental Pathology and Therapeutics Group, Research Center of IPO Porto (CI-IPOP), RISE@CI-IPOP (Health Research Network), Portuguese Oncology Institute of Porto (IPO Porto), Porto Comprehensive Cancer Center (Porto.CCC), 4200-072 Porto, Portugal; daniela.r.santos@ipoporto.min-saude.pt (D.A.R.S.); i12812@ipoporto.min-saude.pt (C.G.); lucio.santos@ipoporto.min-saude.pt (L.S.); 2School of Health, Polytechnic Institute of Porto, Rua Dr. António Bernardino de Almeida, 400, 4200-072 Porto, Portugal; mes@ess.ipp.pt; 3Centro de Investigação em Saúde e Ambiente (CISA), Escola Superior de Saúde, Instituto Politécnico do Porto, 4200-072 Porto, Portugal; 4Molecular Oncology & Viral Pathology, IPO-Porto Research Center (CI-IPO), Portuguese Institute of Oncology, 4200-072 Porto, Portugal; 5Department of Surgical Oncology, Portuguese Institute of Oncology (IPO-Porto), 4200-072 Porto, Portugal; 6Precancerous Lesions and Early Cancer Management Group, Research Center of IPO Porto (CI-IPOP), Rise@CI-IPOP (Health Research Group), Portuguese Institute of Oncology of Porto (IPO Porto), Porto Comprehensive Cancer Center (Porto.CCC), 4200-072 Porto, Portugal; mario.ribeiro@ipoporto.min-saude.pt; 7Department of Gastroenterology, Portuguese Oncology Institute of Porto, 4200-072 Porto, Portugal

**Keywords:** microRNA, colorectal cancer, biomarker, screening, early detection

## Abstract

Colorectal cancer (CRC) ranks as the third most prevalent cancer worldwide. Early detection of this neoplasia has proven to improve prognosis, resulting in a 90% increase in survival. However, available CRC screening methods have limitations, requiring the development of new tools. MicroRNA biomarkers have emerged as a powerful screening tool, as they are highly expressed in CRC patients and easily detectable in several biological samples. While microRNAs are extensively studied in blood samples, recent interest has now arisen in other samples, such as stool samples, where they can be combined with existing screening methods. Among the microRNAs described in the literature, microRNA-21-5p and microRNA-92a-3p and their cluster have demonstrated high potential for early CRC screening. Furthermore, the combination of multiple microRNAs has shown improved performance in CRC detection compared to individual microRNAs. This review aims to assess the available data in the literature on microRNAs as promising biomarkers for early CRC screening, explore their advantages and disadvantages, and discuss the optimal study characteristics for analyzing these biomarkers.

## 1. Introduction

Colorectal cancer (CRC) is the third most common cancer globally, posing a significant public health challenge with far-reaching societal implications [1,2]. The progression of CRC follows a multistage process, transitioning from benign adenomas to malignant adenocarcinomas and eventually spreading to distant sites through metastasis. This intricate progression is influenced by genetic and epigenetic alterations in several molecules associated with cellular and signaling pathways related to CRC (Figure 1) [3,4]. The slow development of CRC makes it particularly suitable for screening, offering a crucial opportunity for early detection, as the disease often manifests without noticeable symptoms [1,2,5]. In fact, the stage at which CRC is detected directly correlates with the disease prognosis, underscoring the significance of timely identification [6,7]. Early detection, specifically in stages I and II, can lead to a remarkable 90% 5-year survival for affected individuals (40% of patients). In turn, 60% of CRC cases remains undiagnosed until advanced stages (stage III–IV), resulting in unfavorable prognosis and increased mortality rates. Therefore, implementing early-stage screening represents a highly effective approach to reduce the burden of CRC [5,8,9].

Current CRC screening methods can be broadly categorized into two main approaches: stool tests to detect occult blood and endoscopic exams, such as colonoscopy [7,10].

Stool tests represent the initial non-invasive strategy for CRC screening [6,7]. Among these, the Fecal Immunochemical Test (FIT) has emerged as the most widely adopted worldwide, replacing the guaiac fecal occult blood test (gFOBT) due to its advantages, such as not requiring dietary restrictions and exhibiting higher sensitivity and specificity in detecting CRC [5,11,12].

FIT demonstrates higher sensitivity (approximately 70%) in detecting malignant lesions, such as adenocarcinomas [8]. However, its sensitivity for detecting precancerous lesions, which may bleed sporadically, is only around 30%, resulting in a significant number of false negatives, thus restricting its overall effectiveness [8,10]. Furthermore, FIT’s low specificity leads to a high false positive rate, requiring unnecessary colonoscopies and subsequently increasing healthcare costs [2,12].

Colonoscopy is widely regarded as the gold standard method for CRC screening, offering a high sensitivity (over 88%) for detecting precancerous and cancerous lesions. Additionally, it allows the removal of adenomas, effectively reducing the incidence of CRC [6,8,9,10]. Nonetheless, colonoscopy is an invasive procedure with inherent limitations, including the risk of complications (e.g., perforation, bleeding, and the risk of adverse reactions to anesthesia), the necessity for bowel preparation, the requirement for anesthesia, poor patient adherence, and high cost [2,5,13,14]. While colonoscopy remains pivotal for diagnostic purposes, its feasibility as a population-wide screening option is hampered by these limitations. Coupled with the limitations of FIT, the early diagnosis of CRC continues to pose significant challenges nowadays [5].

In an effort to address the limitations of existing screening methods, two biomarker-based screening tools, namely, Cologuard^®^ and Epi proColon^®^, have recently emerged as non-invasive approaches for improving the early detection of pre- and cancerous lesions [15,16]. Cologuard^®^ (Multitarget stool DNA testing) is a fecal DNA analysis that identifies biomarkers based on DNA and occult blood. It detects abnormal levels of DNA and/or blood associated with precancerous or cancerous lesions, prompting the patient to undergo a diagnostic colonoscopy [8]. While Cologuard^®^ demonstrated higher sensitivity than FIT in identifying CRC (92.3%) and advanced adenomas (42.3%), its ability to distinguish precancerous advanced lesions from normal tissue remains limited [5,10]. DNA biomarkers can also be found in blood, including DNA methylation [5]. The Epi proColon^®^ test relies on detecting SEPT9 methylation, which is associated with early stages of CRC [17]. While Epi proColon^®^ exhibits the same sensitivity as FIT for CRC diagnosis (70%) and higher sensitivity for symptomatic CRC (77%), its sensitivity for distinguishing advanced adenomas from normal tissue is lower than FIT (approximately 11%), indicating that it is only superior to FIT for symptomatic patients [5,18]. Moreover, this biomarker is associated with cancers other than CRC, such as breast and prostate cancer, increasing the rate of false positives [17]. Despite the sensitivity and specificity of these tests in detecting precancerous lesions, they possess additional limitations, including cost, accessibility challenges, and lower patient adherence [5]. Figure 2 illustrates the different methods for CRC detection.

The pressing need to develop a non-invasive screening method that surpasses the limitations of FIT and colonoscopy is evident, considering factors such as improved sensitivity and specificity, reduced risks of complications and associated costs, and the ability to facilitate early detection of precancerous lesions and advanced adenomas. Such advancements would not only lead to enhanced healthcare outcomes, but also potentially alleviate expenses related to advanced cancer treatment [19]. In the past few decades, several molecules have been investigated as potential biomarkers to enhance CRC screening, and among them, microRNAs have emerged as one of the most promising candidates to overcome the limitations of current methods and significantly improve early detection of CRC [20,21,22].

MicroRNAs are small RNA molecules naturally present in cells, typically ranging from approximately 18 to 24 nucleotides in length [23,24,25]. They play a crucial role in post-transcriptional gene regulation by binding to the 3′ untranslated region of target mRNA, thereby causing its degradation or inhibiting translation. MicroRNAs exert negative regulation over the expression of at least 30% of all protein-coding genes, impacting a wide array of genetic and cellular processes, including proliferation, development, differentiation, apoptosis, inflammation, and stress response [23,24,26]. Moreover, these non-coding RNAs play a significant role in cancer pathogenesis as either oncogenic or tumor suppressor molecules. Their expression levels undergo alterations throughout different stages of cancer progression, making microRNAs a promising biomarker approach for cancer diagnosis and prognosis [27,28,29,30,31,32].

MicroRNAs offer distinct advantages over other biomarkers, as they are intricately involved in cancer mechanisms, exhibit specific up- and downregulation profiles during tumor development, and can be correlated with treatment response and patient survival [25,33]. Furthermore, they can be detected in various body fluids, including blood and stool, making them ideal candidates for CRC screening, enabling the differentiation of individuals with cancer or advanced adenoma from healthy individuals [20,27,28,34]. Exploiting these advantages, microRNA biomarkers in stool samples have the potential to serve as a novel, noninvasive method for CRC screening, particularly when used in conjunction with FIT [35,36].

The objective of this review is to explore and elucidate the potential role of microRNAs in CRC screening. We aim to conduct a thorough and comprehensive review of the existing literature, with a specific focus on studies that evaluate microRNAs as potential biomarkers for the early detection of precancerous lesions in CRC. Our goal is to identify and highlight the most promising microRNAs that exhibit potential as effective diagnostic indicators. Additionally, we will discuss the optimal study designs and methodologies that should be employed when analyzing these microRNA biomarkers, taking into account their unique characteristics and potential clinical applications in CRC screening.

## 2. Analyzing Study Design for microRNA Biomarkers in CRC Screening

MicroRNAs have been the subject of extensive research in various types of cancer and other diseases [37,38]. In the case of CRC, a growing body of evidence suggests that these biomarkers hold potential for application in screening and could improve the current diagnostic methods [39,40]. However, to determine the viability of specific microRNAs as a promising biomarker for CRC detection, a critical analysis of the study design and its characteristics is essential. Several aspects of study design play a pivotal role in evaluating the performance of a given biomarker in the context of CRC screening.

In this comprehensive review, a literature search was conducted using the PUBMED and Science Direct databases. The search included the following keywords: “tumor biomarkers”, “colorectal cancer”, “microRNA”, “stool samples”, “circulating microRNA”, “blood samples”, and “early screening”. A total of 54 reports, published between 2010 and 2023, in the English language, with available abstracts or full text were included. These articles described the altered expression of microRNAs in CRC screening.

This section will examine important study design characteristics that significantly impact the assessment of a biomarker’s performance in CRC screening. Specifically, we will discuss the selection of biological samples used in the evaluated studies, the inclusion of diverse comparison groups, the determination of an appropriate sample size, and the methodology employed for microRNA selection. By thoroughly evaluating these aspects, we can gain a comprehensive understanding of the strengths and limitations of the studies and their implications for the potential use of microRNAs as effective biomarkers in CRC screening.

### 2.1. Biological Samples

MicroRNAs have been detected in various biological samples, such as blood, stool, tissue, saliva, and urine [20,41,42,43]. These biomarkers have shown higher expression in tumor tissues compared to normal tissue and can be easily detected in body fluids [29,44,45]. In the context of CRC, microRNA alterations are predominantly observed in stool and blood samples (circulating microRNAs), which have been extensively studied and reported in the literature (Figure 3). The abundance of research focusing on these samples is attributed to their ability to reflect the microRNA levels released into the blood, colon, and rectum [25,28,29].

Blood samples have been the central focus of microRNA studies in scientific literature, with a significant majority of research (65%—37 out of 54 articles) dedicated to this specific sample type. In recent years, emerging evidence has demonstrated that microRNAs are secreted into the bloodstream and play a crucial role in the exchange of genetic information between tumor cells, contributing to their proliferation and development [46]. This discovery opens up possibilities for using serum and plasma microRNAs in the identification of CRC or advanced adenoma, as their levels are elevated due to their involvement in the tumorigenesis process of CRC [47]. Circulating microRNAs exhibit a high level of stability, enabling their easy detection using techniques such as real-time PCR [48,49]. These small RNA molecules have been shown to be more resilient than mRNAs, remaining stable at room temperature for up to 24 h and exhibiting resistance to extreme pH and temperature conditions [50].

MicroRNAs within blood samples have been extensively studied, with a particular focus on serum samples (23 studies analyzed serum samples, while 14 studies used plasma specimens). The behavior of microRNAs in serum samples is better understood. In serum, circulating microRNAs are either encapsulated in exosomes or microvesicles or transported by proteins, which confer high stability to these small molecules, protecting them from degradation by ribonucleases [51,52]. In plasma, microRNAs are released into the extracellular space by colon cells during processes such as tumor proliferation or cell apoptosis. However, it is important to note that plasma also contains microRNAs derived from blood cells, which can potentially contaminate the samples [53]. Moreover, compared to plasma, serum samples are less complex to process, and the microRNAs are less exposed to high levels of ribonucleases present in plasma [50,54].

After analyzing the literature, it can be concluded that the detection of microRNAs in blood, specifically in serum samples, is a feasible and robust method for improving CRC screening. Furthermore, they can be combined with other existing methods, such as FIT, to provide a less invasive option, which is advantageous for patients [47,55].

In addition to blood, microRNAs are also commonly found in stool samples and can be detected in patients with CRC or colorectal adenomas [56,57,58]. These short molecules are highly expressed in stool samples due to the continuous shedding of colon cells in the gastrointestinal tract [56]. This ongoing shedding theoretically contains genetic and epigenetic information relevant to carcinogenesis, including dysregulated mRNA, proteins, and microRNAs [47]. Depending on the stage of CRC, colon tumor cells release varying levels of a plethora of microRNAs, which can be directly detected in the feces, reflecting the actual characteristics of CRC lesions and allowing early detection of precancerous lesions [28,56]. Studies involving stool samples suggest that microRNAs can identify early lesions of CRC with a diameter of less than 1 cm and offering higher sensitivity compared to FIT in cancer screening [20,47,55]. The detection of microRNAs in stool samples provides a promising approach for early CRC detection, leveraging their ability to capture the molecular signatures of the disease.

Although microRNAs have been extensively studied in serum, evaluating microRNAs present in stool samples may offer more advantages. Fecal microRNAs, compared to circulating microRNAs, exhibit greater stability, remaining intact for up to 72 h at room temperature and resisting degradation by ribonucleases [25,26]. Despite the presence of potential interfering factors, such as food and intestinal bacteria in stool samples, the utilization of highly reproducible quantification methods can effectively overcome these challenges [47,48]. From the patients’ perspective, collecting and storing fecal samples is easy, providing a noninvasive screening method [28]. Moreover, some studies suggest that these short molecules in stool samples remain stable in FOBT buffer for up to 5 days when stored at 4 °C, allowing for their analysis in conjunction with FIT [20,40].

Thus, after evaluating the advantages of fecal and circulating microRNAs, it can be concluded that while both sample types offer important benefits, stool samples should be more widely explored due to their close proximity to colon cells. By harnessing the unique advantages of fecal microRNAs, we can enhance our understanding and detection of colorectal cancer and its precursors.

### 2.2. Study Groups

The precise definition of the comparison groups is essential for evaluating the specificity and predictive value of microRNA biomarkers for early detection of CRC.

To assess the potential of microRNA as noninvasive screening biomarkers, the studies included both positive and negative controls. In all reports, the positive control group consisted of recently diagnosed CRC patients who had not undergone any cancer treatment, ensuring that their microRNAs levels remained unchanged. However, the composition of the negative control group varied among the studies. Approximately 60% of the articles (32 out of 54 studies) recruited healthy individuals who had undergone colonoscopy, which did not reveal any lesions, despite having either a negative or positive FIT result. On the other hand, 40% of the reports (22 out of 54 articles) enrolled patients with a negative FIT result or individuals with no history of cancer. It is important to acknowledge the limitations associated with the inclusion of these patients. Firstly, the high false negative rate of FIT should be taken into consideration. Secondly, it is worth noting that even healthy individuals without a history of cancer may have asymptomatic lesions. Consequently, the presence of individuals in the negative control group who potentially have undetected lesions could impact the results due to differences in microRNA expression patterns.

In addition to the aforementioned control groups, 46% of the articles (25 out of 54 reports) included additional patient groups with premalignant lesions, such as colorectal adenoma and advanced adenomas. The inclusion of these groups is crucial as it allows the evaluation of microRNAs expression across the several stages of CRC development, ranging from normal to malignant phenotypes. This comprehensive assessment enables the determination of the sensitivity and specificity of the biomarker in detecting early lesions, which is a major limitation of the current screening methodologies. However, out of the 54 articles reviewed, only 11 (20%) included all these comparison groups, thereby allowing a proper assessment of their ability to accurately detect both cancerous and premalignant lesions.

Therefore, it is relevant to employ a study design that includes diverse comparison groups, encompassing patients without lesions in the colon and rectum, patients with colorectal adenoma, advanced adenoma, and CRC. These groups represent the multi-step progression of CRC, facilitating the determination of the microRNA sensitivity in detecting precancerous lesions where FIT has poor detection capabilities. This approach provides a significant advantage over other screening methods [8].

### 2.3. microRNA Selection

Once the study groups have been established, it becomes critical to determine the selection of microRNAs for analysis.

In our search, the selection of microRNAs varied among the studies: 38% (21 out of 54 articles) did not explicitly mention how microRNAs were selected; 24% (13 out of 54 studies) employed a candidate-based approach, selecting microRNAs based on previous findings in the literature; 20% (11 out of 54 reports) applied in silico analysis of public databases; and 17% (9 out of 54 studies) employed high-throughput approaches, such as Next Generation Sequencing (NGS) and microarrays.

Each of these approaches possesses its own advantages and limitations, which will be explored below. The high-throughput approach offers the advantage of evaluating hundreds of microRNAs simultaneously, increasing the likelihood of identifying microRNAs with significant differential expression profiles that may have otherwise remained unexplored. However, it is important to note that this method can be costly [59,60]. On the other hand, literature review and bioinformatics are low-cost methods for identifying microRNAs but are constrained by existing knowledge. Bioinformatics also allows for rapid identification of microRNA profiles based on previous data; however, experimental validation is necessary to fully comprehend the microRNA expression levels in biological samples. Meanwhile, literature review is a straightforward procedure for identifying microRNAs. It is particularly useful in determining potential combinations of microRNAs that have not been previously tested and allows for predictions regarding the behavior of the biomarker in the sample, anticipating certain results.

Considering the advantages and limitations of different microRNA selection methods, it can be deduced that the utilization of high-throughput techniques presents the optimal approach for selecting these biomarkers. These techniques allow for a comprehensive analysis of microRNA expression in the specific sample under evaluation. Notably, approaches such as RNA-sequencing have the potential to reveal previously unexplored biomarkers that have not been assessed before [60]. Nevertheless, it is worth noting that the candidate-based approach, which relies on the identification of predetermined targets described in the literature, remains the most commonly employed method.

### 2.4. Sample Size and Patients Recruitment

Another critical aspect to consider in study design is the sample size. In our review, the sample sizes varied, ranging from 28 to 767 patients, with an average of 70 patients for validation studies and 225 patients for articles with a biomarker confirmation phase [20,22]. Within these patient populations, the number of healthy controls (including patients who underwent colonoscopy without any lesions detected) ranged from 10 to 217. For patients with colorectal adenoma, the range was from 9 to 164, and for patients with advanced adenoma, the range was from 9 to 347. The number of patients diagnosed with CRC included in the studies also varied, ranging from 10 to 307. The inclusion of a substantial cohort of patients with precancerous lesions in some studies is particularly noteworthy, as mentioned earlier. This inclusion enables the assessment of biomarker accuracy in addressing the primary limitation of current standard methods for CRC screening.

In addition to sample size, patient’s recruitment site is another crucial factor to consider. The majority of the analyzed articles recruited patients from hospitals, specifically those with mandatory positive FIT results. However, one study stood out by recruiting patients from a screening program [20]. Recruiting patients from screening programs offers the advantage of being more representative of the general population, reducing bias risk. Additionally, this approach enables the evaluation of the potential biomarkers in early detection of CRC, especially in stages I and II, as well as advanced adenoma. Moreover, including FIT-positive patients without lesions on confirmatory colonoscopy allows a more accurate evaluation of biomarker performance, since this group encompasses the false positive results of FIT. On the other hand, when recruiting patients in hospital settings, a higher proportion of advanced stage CRC cases is often found, specifically stages III and IV, and normally lacks FIT negative patients [36].

In conclusion, recruitment only from hospitals could be limiting. Moreover, the recruitment from screening programs, including both FIT positive and negative patients brings more advantages, providing the opportunity to accurately define the performance of the biomarker. Additionally, researchers must calculate the adequate sample sizes to ensure a robust statistical power to assess the screening value of microRNAs.

## 3. Promising microRNAs: Potential Biomarkers for CRC

MicroRNAs have been widely recognized in scientific literature as promising candidates for cancer screening [31]. These small noncoding RNAs contain multiple target genes and play a significant role in regulating cancer-related pathways [37]. In CRC, the dysregulated expression of microRNAs is associated with different clinical stages, enabling the assessment of cancer progression [26,27]. The altered expression of microRNAs can be detected in biological samples, such as feces and blood, offering a more accurate and non-invasive method that brings advantages for the patients. The use of microRNA presents a unique opportunity for the early detection of CRC [27,61,62,63,64,65].

Through a comprehensive literature review, a total of 104 microRNAs were evaluated across 54 articles, focusing on stool and/or blood samples (Appendix A). Among these microRNAs, 65% (68 microRNAs) were described in blood (60 in serum and 57 in plasma), 10% (11 microRNAs) were identified in feces, and 25% (25 microRNAs) were present in both sample types. The reported sensitivity, specificity, and Area Under the Curve (AUC) of microRNAs for CRC screening varied across the studies, ranging from 14.70% to 98.00%, 47.00% to 100.00%, and 0.525 to 0.994, respectively. Notably, circulating microRNAs exhibited the most favorable values [20,66,67,68,69].

Among the 104 microRNAs mentioned in the literature, a subset of 60 microRNAs (57%) have been described to be associated with the detection of malignant or pre-malignant lesions, underscoring their potential as biomarkers for CRC detection and their ability to augment screening efforts (Figure 4). Among these 60 microRNAs, 68% (41 microRNAs) were analyzed in blood samples, 10% (6 microRNAs) in stool samples, and 22% (13 microRNAs) were detectable in both specimen types.

These microRNAs demonstrate an AUC higher than 0.700, with sensitivity and specificity values exceeding 60% in detecting CRC. Furthermore, their analysis has demonstrated reproducibility and strong association with the pathogenesis of CRC, indicating altered expression during cancer progression [20,56,70,71]. In the subsequent section, we provide a comprehensive overview of the microRNAs with exceptional performance described in the literature (sensitivity and specificity above 80%), all of which are represented in Figure 5.

### 3.1. Comprehensive Analysis of Highly Described microRNAs

#### 3.1.1. microRNA-21-5p

MicroRNA-21-5p, one of the earliest microRNAs discovered in humans, has garnered significant attention in the literature [26,61]. With 17 studies dedicated to its exploration, microRNA-21-5p emerges as a promising noninvasive biomarker for CRC detection, as demonstrated by 14 of these studies. Its effectiveness lies in its remarkable sensitivity and specificity in identifying CRC. In stool samples, sensitivity ranges from 14.70% to 88.10% (mean: 71.74%), while in blood samples, it ranges from 54.70% to 90.63% (mean: 76.10%). Specificity varies from 48.30% to 95.00% (mean: 71.50%) in stool samples, and from 70.60% to 100.00% (mean: 84.40%) in blood specimens [20,27,31,61,66,67,72]. This oncogenic microRNA exerts its influence through the modulation of multiple cancer target genes and regulation of pathways involved in tumorigenesis [61,66,67,73]. It also plays a role in apoptosis, cell proliferation, cell migration, and angiogenesis by targeting genes, such as phosphatase and tensin homolog (*PTEN*), programmed cell death protein 4 (*PDCD4*), protein sprouty homolog 1 (*SPRY1*), and nuclear factor 1B [31,72,74,75]. Upregulated expression of microRNA-21-5p has been observed in CRC tissues and can be detected in noninvasive samples, such as blood and stool. Its upregulation is influenced by both genetic and epigenetic alterations, impacting tumor progression and showing associations with TNM staging, poor prognosis, and overall survival [26,61,75]. Moreover, overexpression of microRNA-21-5p significantly enhances resistance to 5-FU and radiation in CRC tumors, highlighting its potential as both a diagnostic biomarker and a tool for assessing therapy response [76]. Additionally, microRNA-21-5p has been linked to other cancers, including pancreatic, breast, lung, and gastric cancers [77,78,79,80,81]. Its widespread significance across multiple cancer types underscores its potential as a versatile target for diagnostic and therapeutic approaches.

#### 3.1.2. microRNA-92a-3p and Cluster miR-17-92

MicroRNA-92a-3p has emerged as a highly evaluated microRNA in CRC screening research, with 13 studies dedicated to its investigation. Impressively, 62% of these studies (8 articles) have demonstrated its potential for early-stage CRC detection. Sensitivity ranges from 15.70% to 89.70% (mean: 64.80%) in stool samples and from 65.50% to 84.00% (mean: 74.80%) in blood samples, while specificity ranges from 51.70% to 90.80% (mean: 74.80%) in stool samples and from 71.20% to 82.50% (mean: 81.50%) in blood samples [27,67,82,83]. Functioning as an oncogene, microRNA-92a-3p induces epithelial-to-mesenchymal transition and regulates cell growth, migration, and invasion by suppressing PTEN expression [27,29,84]. In CRC, this microRNA is consistently upregulated, and elevated levels have been correlated with TNM staging and a poor prognosis [27]. Notably, combining microRNA-92a-3p with microRNA-21-5p and other microRNAs has shown increased effectiveness in detecting CRC and precancerous lesions. Furthermore, microRNA-92a-3p, such as microRNA-21-5p, is also under investigation in other cancers, including gastric and breast cancer [85,86]. MicroRNA-92a-3p is a member of the miR-17-92a cluster, which encompasses several other oncogenic microRNAs, such as microRNA-17-5p, -18a-5p, -19a-3p, -19b-3p, -20a-5p, and -92a-3p [87,88,89]. Extensive research has focused on microRNAs from the miR-17-92a cluster in the context of CRC screening, with their detection demonstrated in both stool and blood samples. The cluster has been observed to be overexpressed in CRC and plays a role in disease development processes, while also predicting therapeutic response [88,89,90]. Among the cluster members, microRNA-18a-5p was the third most extensively studied microRNA (nine articles) and proved useful for CRC detection in five reports. MicroRNA-29a-3p and microRNA-20a-5p were described in eight and seven studies, respectively, and were considered potentially valuable for CRC screening in four articles. MicroRNA-19a-3p, microRNA-19b-3p, and microRNA-17-5p were less explored in comparison. Although most cluster members, excluding microRNA-92a-3p, did not exhibit strong performance in detecting cancer, they were still considered potentially useful for CRC screening in 7 out of 23 studies. It is important to note, however, that the majority of these articles only provided the AUC value (ranging from 0.600 to 0.929, with higher values observed in blood samples), without reference to sensitivity and specificity values [91,92,93]. This reliance on the AUC value alone hinders a direct comparison with other methods used for CRC detection and leaves the interpretation of the microRNA’s ability to distinguish between CRC and healthy patients solely based on the AUC value. Nevertheless, the remaining studies that reported sensitivity and specificity values enable us to draw some conclusions. In stool samples, the specificity of all microRNAs within the cluster is higher than that of FIT, but the sensitivity is lower, except for microRNA-29a-3p and microRNA-92a-3p. On the other hand, in blood samples, the latter two microRNAs, along with microRNA-18a-5p, perform just as well as in stool, with higher sensitivity values [67,82,94,95]. Therefore, microRNA-92a-3p emerges as a promising biomarker for early-stage CRC detection, exhibiting elevated expression levels in both stool and blood samples. Additionally, microRNA-29a-3p and microRNA-18a-5p, two members of the miR-17-92a cluster, have shown interesting results, particularly in plasma samples. However, further validation in other types of specimens is necessary to establish their reliability and clinical significance.

#### 3.1.3. Exploring Other Key microRNAs in CRC Screening

Several other microRNAs have been extensively studied in blood and stool samples in the available literature, demonstrating their potential for enhancing CRC screening. 

Notably, microRNA-27a-3p has been described in four studies, with three of them showing encouraging results. Overall, this small RNA molecule exhibited better performance in blood samples compared to stool samples [20,96,97]. Specifically, in blood samples, it demonstrated high sensitivity values in plasma (75.00% and 81.82%) [96,97]. However, in serum samples, it could not be considered a potential biomarker for CRC screening, as it exhibited low sensitivity (approximately 43%), despite a relatively high specificity value (88.57%) [74]. Further studies are needed to validate the diagnostic potential of microRNA-27a-3p in different types of specimens, particularly in stool and serum samples.

Another microRNA that has shown favorable results in three studies for the detection of CRC in serum samples is oncogene microRNA-23a-3p. It is also one of the microRNAs with the highest sensitivity and specificity values in serum samples (91.00% and 91.34%, respectively) [41,74,98]. Based on existing literature, increased levels of microRNA-23a-3p are associated with advanced stages of CRC, as its overexpression suppresses the apoptotic process by repressing the expression of pyruvate dehydrogenase kinase 4 and apoptosis-activating factor-1 (*APAF-1*) [74,99]. Although microRNA-23a-3p is one of the most promising microRNAs in serum samples, it has not been evaluated in stool samples, requiring further validation in this biofluid. MicroRNA-146a-5p is an intriguing biomarker, exhibiting potential value, as evidenced by two reports that assessed both stool and blood samples [66,73,82]. This microRNA showed favorable sensitivity and specificity values for CRC detection in stool (77.20% and 68.10%, respectively) and even more promising results in serum samples (98.00% and 74.10%, respectively) [66,73]. However, the role of this microRNA in CRC remains controversial [100]. El Din et al. reported upregulation of microRNA-146a-5p, while Liu et al. suggested downregulation [66,73]. By contrast, Huang et al. found no significant difference in the expression levels of this microRNA between CRC patients and healthy individuals [82]. These discrepancies in the expression of microRNA-146a-5p may be attributed to variations in sample types, RNA extraction and quantification methods, control types, and patient heterogeneity [41,47,101,102,103]. Despite the inconsistencies observed between the two sample types, microRNA-146a-5p has demonstrated utility in both stool and serum samples, warranting further validation of these findings in both sample types. MicroRNA-135b-5p arises as another promising biomarker for CRC screening, particularly in stool samples. It has been evaluated in five studies, and three of them have reported compelling results [28,57,67]. Li et al. reported a remarkable sensitivity of 96.50% and specificity of 74.10% for microRNA-135b-5p in a cohort of 106 patients [28]. Significantly, this represents the highest sensitivity value described for a microRNA in stool samples, despite the relatively small sample size. In contrast, Wu et al. analyzed this microRNA in a larger cohort (n = 424) across various stages of CRC and found relatively lower sensitivity and specificity values (78.00% and 68.00%, respectively). Similar performance was observed for microRNA-1290 in three studies conducted in blood samples. It demonstrated the most favorable outcomes in serum samples in a cohort of 135 patients, but the values decreased when evaluated in a larger cohort of patients (n = 324), with a sensitivity of 83.33% reported in the study with 135 patients and 70.10% sensitivity in the larger cohort [45,57,98]. Subsequently, microRNA-223-3p and microRNA-221-3p are two extensively studied microRNAs in the context of CRC screening. Regrettably, studies focusing on blood samples only reported AUC values, without providing information on sensitivity and specificity [41,72,82,104]. In stool samples, microRNA-223-3p showed a performance similar to the previously discussed microRNAs, showing approximately 70.00% sensitivity and 80.00% specificity. However, there was a decrease in both sensitivity and specificity as the number of patients increased [44,105]. Conversely, microRNA-221-3p, evaluated in stool samples, was examined in 2 large-scale studies with patient cohorts of either 595 or 767, demonstrating, in 1 study, superior sensitivity (70.00%) compared to FIT for detecting advanced adenomas (30.00%) [20,106].

While a significant number of the microRNAs identified in our search have been deeply evaluated in several studies and exhibited good accuracy in identifying CRC, we have also come across other microRNAs, such as microRNA-143-5p, microRNA-145-5p, microRNA-106a-5p, microRNA-106b-5p, microRNA-181d-5p, microRNA-31-5p, and microRNA-16-5p, that did not exhibit the same potential for CRC screening. For instance, microRNA-31-5p, despite being explored in blood samples by six reports, has not shown altered levels in CRC. However, when evaluated in tissue samples, this microRNA appears to hold promise, particularly in early-onset detection of CRC [107]. Another example is microRNA-16-5p, which has been evaluated in two studies involving both stool and serum samples, yet no information about sensitivity and specificity values was provided. Nevertheless, recent evidence suggests that microRNA-16-5p can also serve as a reference gene in CRC, given its stable expression observed in certain studies [102,108]. Thus, numerous microRNAs have exhibited potential as biomarkers for CRC detection, displaying varying sensitivities and specificities depending on the type of specimen analyzed. Furthermore, for some microRNAs, additional studies are needed to report sensitivity and specificity values, in addition to AUC values, to provide a more comprehensive understanding of their diagnostic benefit.

### 3.2. microRNAs with Encouraging Prospects Requiring Further Validation

Although several microRNAs have been widely studied in both stool and blood samples, our search has unveiled a set of less explored microRNAs showing promising outcomes. Whitin this group, we have identified a total of 78 microRNAs, mostly in blood samples, among which 40 demonstrate some degree of potential in identifying individuals with CRC. In this section, we will focus on microRNAs that have exhibited superior performance in detecting CRC patients.

Among the analyzed biomarkers, microRNA-139-3p arises as the microRNA displaying the most outstanding performance in serum samples. It showed the highest values of sensitivity, specificity, and AUC in our investigation, with rates of 96.60%, 97.80%, and 0.994, respectively, in a cohort of 207 patients [68]. Notably, this microRNA is known to be downregulated not only in CRC, but also in other cancer types, such as hepatocellular carcinoma, gastric cancer, breast cancer, and ovarian cancer [68,109,110,111,112]. Conversely, its overexpression in healthy individuals helps in suppressing tumor cell proliferation and invasion [110]. Although the precise role of microRNA-139-3p in CRC progression remains unclear, one study has suggested its role as a tumor suppressor, correlating it with disease severity and poor patient survival [110]. Further studies are necessary to fully understand the mechanisms underlying the action of microRNA-139-3p in CRC and confirm its accuracy as a biomarker for CRC screening. In addition, microRNA-4516 has been identified as an emerging biomarker that demonstrates impressive efficacy in serum samples. This molecule is downregulated in CRC and has been associated with TNM staging, invasion, and metastasis [61]. However, the precise underlying mechanisms by which it influences CRC progression are yet to be fully elucidated [61]. In a study involving 195 patients, microRNA-4516 revealed high sensitivity, specificity, and AUC values for CRC screening, with rates of 94.40%, 89.80%, and 0.944, respectively. Nonetheless, further evaluation and validation in other biological fluids are required to assess its potential as a diagnostic biomarker for CRC [61]. Moreover, in blood samples, several other microRNAs have shown promising results, with high sensitivity values ranging from 80 to 90%. These microRNAs include microRNA-144-3p, -199-3p, -423-5p, -320-3p, -940, -627-5p, -592, and -760, which have been analyzed in studies encompassing more than 100 patients [70,96,98,113,114]. Regarding fecal specimens, microRNA-451a has displayed high sensitivity, specificity, and AUC values of 88.00%, 100.00%, and 0.971, respectively [44]. Although this microRNA has also been evaluated in serum samples, its association with CRC identification was only observed in stool samples. In serum samples, the AUC value for microRNA-451a was 0.650, indicating a poorer performance compared to fecal specimens [91]. However, it is worth noting that the studies assessing this microRNA had small patient cohorts, which might have influenced the obtained values. Therefore, validation of this microRNA in a larger patient cohort and using both biological samples is necessary.

MicroRNA-144-5p has showcased favorable outcomes for CRC detection, although its assessment has been confined to stool samples, where it displayed a sensitivity of 78.90% and a specificity of 87.20% [27]. On the other hand, microRNA-144-3p, derived from the -3p arm of the hairpin and showing higher expression, unveiled a higher sensitivity value of 93.80% in plasma samples [96]. Hence, it would be interesting to analyze the performance of this microRNA in stool samples or even investigate the combined expression of both microRNAs. 

In summary, although several microRNAs have shown encouraging outcomes, further additional studies are imperative to establish their validity as reliable biomarkers for early detection of CRC. An essential factor to take into account is the inclusion of individuals with advanced adenomas, as it allows for the assessment of microRNAs’ accuracy in detecting precancerous lesions. This aspect holds particular significance since the majority of microRNAs have primarily been evaluated in distinguishing between healthy individuals and those diagnosed with CRC. Therefore, further research addressing these considerations is essential to advance the understanding and the utility of microRNAs in CRC screening.

### 3.3. Potential microRNAs for Precancerous Lesions Identifcation

MicroRNAs offer distinct advantages in the early detection of CRC due to their involvement in several signaling pathways [71]. A plethora of microRNAs, namely, microRNA-21-5p, microRNA-146a, microRNA-135b-5p, microRNA-29a-3p, microRNA-1290, microRNA-592, microRNA-601, microRNA-760, microRNA-130a-3p, microRNA-627-5p, microRNA-199a-5p, microRNA-421, microRNA-130b-3p, microRNA-335-3p, microRNA-34a-5p, and microRNA-27a-3p, have been exhaustively studied for their potential in early diagnosis, focusing on their ability to detect premalignant lesions [20,45,57,73,82,113,115,116].

In a study by Liu et al., microRNA-21-5p and microRNA-146a-5p demonstrated high sensitivity in stool samples for detecting colorectal adenomas, including advanced adenomas (microRNA-21-5p: 72.00% and 85.10% of sensitivity for advanced adenoma and colorectal adenoma detection, respectively; microRNA-146a-5p: 77.50% of sensitivity for colorectal adenoma detection) [20,73]. Huang et al. assessed the efficacy of microRNA-92a-3p in blood samples and found a sensitivity of 64.60% for detecting advanced adenomas, surpassing that of FIT [82]. Duran-Sanchon et al. explored the potential of several microRNAs in stool samples for detecting advanced adenomas and identified microRNAs-130b-3p and -421, both showing sensitivities over 80% for detecting this type of lesion (82.00% and 81.00% respectively). Remarkably, the specificity of these microRNAs was comparable to the FIT test [20]. Additionally, microRNA-199a-5p displayed a sensitivity of 90.00% for advanced adenoma detection in serum samples, positioning it as one of the most promising biomarkers for detecting premalignant lesions [113].

Lastly, microRNA-29a-5p, in particular, showed a higher sensitivity for the detection of advanced adenoma than for CRC in plasma samples, with a sensitivity of 73.00% for advanced adenoma and 69.00% for CRC, as reported by Huang et al. However, its performance in stool samples was not impressive, with a sensitivity of 68.00% for advanced adenoma and 85.00% for CRC [82]. On the other hand, microRNA-135b-5p, strictly investigated for detecting various stages of CRC, particularly in stool samples, has shown to be a reliable biomarker for identifying CRC, advanced adenoma, and colorectal adenomas. It has demonstrated sensitivity and specificity values exceeding 60.00% [57].

Despite the impressive findings of individual microRNAs, their evaluation has certain limitations. As mentioned above, microRNAs are involved in multiple cancers and other diseases, which can affect their specificity for CRC detection. Additionally, the expression of microRNAs in samples, such as stool and blood from CRC patients, could be influenced by alterations in immune response associated with tumor progression [73,75]. To overcome these limitations, the combination of different microRNAs has been proposed as a strategy to improve CRC screening. The following section will discuss the advantages of this approach and highlight the most effective microRNA combinations reported in the literature.

## 4. Unleashing the Potential of Combined microRNAs in CRC Early Detection

In recent years, there has been a growing interest in exploring panels consisting of combinations of different microRNAs as a more comprehensive approach for cancer detection, especially for colorectal cancer [117,118].

Our detailed search resulted in the identification of 35 articles describing 57 microRNA combinations used in CRC screening (Appendix A). Among these panels, 45 were investigated in blood samples (78%), 11 in stool samples (20%), and 1 comprised microRNAs from both plasma and feces (2%) (Figure 6). The reported microRNA panels exhibited a wide range of sensitivity (57.00% to 96.00%), specificity (37.50% to 100.00%), and AUC values (0.639 to 0.954) [27,66,118,119]. Overall, based on the comprehensive review of the literature, it becomes evident that microRNA panels generally demonstrate superior performance, in terms of both sensitivity and specificity, for the detection of CRC and precancerous lesions compared to individual microRNA evaluation [73,82,120,121,122]. These findings are supported by various studies, which are discussed in detail below.

Similarly, to the microRNAs analyzed in the previous section, microRNA-21-5p (within 16 combinations), microRNA-92a-3p (within 15 combinations), and microRNA-223-3p (within 6 combinations) were the most frequent microRNAs found in combinations. Chang et al. extensively evaluated the latter two microRNAs in both stool and plasma samples, using five different combinations with other microRNAs, including one combination involving both sample types. Importantly, all of these combinations have shown sensitivity higher than 70% for CRC detection. The combination of microRNA-223-3p and microRNA-92a-3p, in both plasma and stool samples, has demonstrated an exceptional sensitivity of 96.80% and a specificity of 75.00%. Consequently, the authors concluded that this method could be a cost-effective and superior alternative to FIT testing for CRC detection [29].

Concerning blood samples, there has been greater emphasis on exploring microRNA combinations as opposed to stool samples. Among the blood samples, the microRNA panel with the highest accuracy for CRC detection consisted of three microRNAs: -144-3p, -425-5p, and -1260b. This panel achieved a sensitivity of 93.80% and specificity of 91.30% for detecting CRC, surpassing the performance of individual microRNAs [96]. Similarly, the combined expression of microRNA-21-5p and microRNA-23a-3p in serum has demonstrated higher sensitivity and specificity for identifying CRC (97.14% and 82.86%, respectively) [74]. Moreover, this enhanced performance was extended to the identification of advanced adenoma and other colorectal lesions. Villanueva et al. reported a panel consisting of 6 microRNAs evaluated in plasma (microRNA-19a-3p, -19b-3p, -15b-5p, -29a-3p, -335-3p, -18a-5p) that exhibited a higher potential for detecting advanced adenoma than CRC (95.00% sensitivity for advanced adenoma and 91.00% for CRC, and 90.00% specificity for both types of lesions) [123]. Additionally, the combination of microRNA-92a-3p with microRNA-29a-3p in plasma samples and microRNA-21-5p in serum samples has shown similar efficacy [82,83]. These findings suggest that analyzing circulating microRNAs in combination has the potential to improve CRC screening and enhance detection accuracy, especially for advanced adenomas.

Regarding fecal samples, there is a growing interest in the use of microRNA combinations; however, the number of studies focusing on this type of sample is relatively limited. Wu et al. conducted a study and found that the combination of microRNA-34b-3p and microRNA-34c-5p exhibited the highest accuracy as a panel for detecting CRC in stool samples [118]. Equally, Liu et al. revealed that combining microRNA-21-5p and microRNA-146a-5p showed higher potential as a biomarker for CRC screening compared to their individual evaluation [73]. Similarly, to blood samples, microRNAs in stool are also being assessed for the detection of precancerous lesions. While the aforementioned combination demonstrated only a slight improvement in specificity for detecting colorectal adenoma compared to individual microRNAs analysis, the sensitivity was found to be higher when microRNAs were evaluated separately (approximately 87.00% of sensitivity for the combination, and 90.30% and 77.20% for microRNA-21-5p and microRNA-146a-5p, respectively). On the other hand, the combination of microRNA-135a-5p and microRNA-135b-5p for detecting advanced adenomas exemplifies a scenario where the individual evaluation of microRNA-135b-5p achieved higher specificity values (68.00%), compared to its combination with microRNA-135a-5p (58.00%) [57]. Overall, using combinations of microRNAs in stool samples holds promise for improving the accuracy of CRC screening. However, there is a lack of research on combined microRNA expression and detection of the advanced adenomas. To address this gap, Duran-Sanchon et al. developed the miRFec algorithm, which takes into account the fecal expression of two microRNAs (microRNA-27a-3p and microRNA-421), along with the hemoglobin concentration, sex, and age of the patients. This algorithm is currently undergoing validation in a large-scale clinical trial, and preliminary data suggest an increase in sensitivity for detecting this type of lesion (sensitivity of 79.00%) [40].

In summary, the analysis of multi-microRNA panels holds the potential to enhance accuracy compared to individual microRNAs alone. This approach allows for the inclusion of microRNAs involved in different signaling pathways, offering a way to overcome the limitations of current CRC screening methodologies, such as FIT.

## 5. Final Considerations

Enhancing CRC screening is of utmost importance given its global prevalence. The current noninvasive method, FIT, has limitations in effectively detecting precancerous lesions. However, using microRNA as a biomarker in biological samples, such as blood and stool, holds great promise for improving the efficiency and convenience of CRC screening.

Our extensive literature search, encompassing 54 studies investigating 104 microRNAs and 57 microRNA combinations in stool and blood samples for early screening of colorectal cancer (CRC), reveals compelling evidence supporting the potential of microRNAs as valuable biomarkers. Stool-based microRNAs offer a particularly promising avenue for effective and non-invasive early cancer detection, while blood-based microRNAs have garnered significant attention in recent years.

To ensure the accurate assessment of microRNA biomarkers for CRC screening, the design of studies plays a critical role. Comprehensive evaluation necessitates the inclusion of diverse patient groups, accounting for variations in disease characteristics and stages. This approach enhances the robustness of the evaluation of microRNA biomarkers and strengthens the reliability of the findings.

Among the extensively investigated microRNAs, microRNA-21-5p and microRNA-92a-3p, along with their cluster members (including microRNA-18a-5p, microRNA29a-3p, and microRNA-20a-5p), emerge as noteworthy candidates, consistently mentioned in the literature. Furthermore, several other microRNAs exhibit differential expression patterns in stool and blood samples, such as microRNA-135b-5p, microRNA-223-3p, and microRNA-451 in stool, as well as microRNA-139-3p and microRNA-4516 in blood samples, suggesting their potential for facilitating CRC detection.

In comparison to FIT, additional microRNAs, such as microRNA-146a-5p, microRNA-199a-5p, microRNA-421, microRNA-27a-3p, and microRNA-221-3p, demonstrate promising results in detecting advanced adenomas. However, the validation of these biomarkers in larger patient cohorts is imperative to establish their clinical utility and reliability.

Compared to alternative biomarkers, certain microRNAs have demonstrated higher sensitivity and specificity. In stool samples, microRNAs have shown superiority over DNA methylation as a diagnostic tool for early CRC detection. Conversely, in blood specimens, other molecular biomarkers, such as long non-coding RNAs (e.g., CCAT1) and protein markers (e.g., Guanylyl Cyclase C, Annexin A2), have been investigated, but their performance has not surpassed that of specific microRNAs (e.g., microRNA-21-5p, microRNA-92a-3p, microRNA-139-3p, microRNA-4516). Although alternative biomarkers, such as DNA methylation and protein markers, have been studied, microRNAs offer unique advantages in terms of stability, regulatory functions, and consistent expression patterns, making them promising candidates for accurate and reliable CRC diagnosis.

Moreover, the combined analysis of microRNAs has emerged as a well-explored approach in this comprehensive review. Notably, microRNA-21-5p and microRNA-92a-3p frequently feature in combination panels, effectively enhancing the performance of various combinations. Additionally, combining microRNAs from the miR-17-92a cluster shows promise for detecting advanced adenomas, particularly in blood samples. Noteworthy in stool sample analysis is the miRFec algorithm, leveraging the differential expression of specific fecal microRNAs in conjunction with FIT, sex, and age. Therefore, the integration of multiple microRNAs holds significant potential as a novel non-invasive tool to improve early disease detection, surmounting the limitations associated with standard screening methods, such as FIT, while simultaneously enhancing precision and efficiency.

The findings presented in this review underscore the value of microRNAs as promising candidates for CRC screening. Nonetheless, further research and extensive validation in larger cohorts are necessary to establish the clinical utility and reliability of these biomarkers, paving the way for their integration into routine clinical practice and contributing to more effective and personalized CRC management strategies. Moreover, the meticulous compilation of information presented in this review can assist researchers in narrowing down targets and selecting microRNAs of interest for the validation of promising biomarkers more accurately. Additionally, this review highlights the importance of study design characteristics, which can contribute to the standardization of future studies.

Herein, we focused the selection of the literature exclusively on microRNAs evaluated in the context of CRC screening, aiming to include the most relevant and influential publications to provide readers with a solid foundation for understanding this topic. However, it is acknowledged that a broader inclusion of studies could have potentially added more valuable information. Although a thorough analysis of a significant number of articles was conducted, it is possible that some important reports may have been inadvertently missed, introducing a potential selection bias.

Nevertheless, it is important to note that the approach employed in presenting the information closely aligns with that of a systematic review. While our selection process may not have followed a strict systematic review protocol, we strived to present a comprehensive and informative overview of the available literature on microRNAs in CRC screening. The inclusion of relevant studies, critical analysis of the findings, and systematic organization of the information aimed to provide readers with a reliable and valuable resource in the field of precision medicine in oncology.

## Figures and Tables

**Figure 1 ijms-24-11023-f001:**
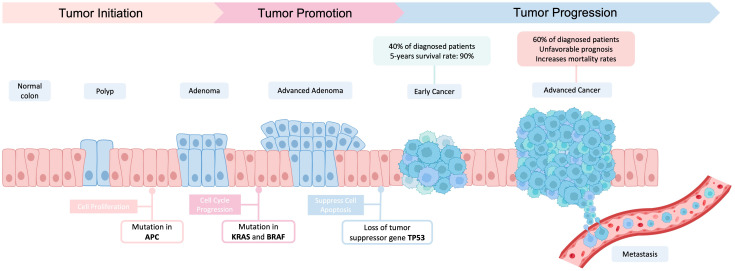
Progression of Colorectal Cancer. Colorectal cancer undergoes a gradual progression marked by multiple steps, encompassing diverse genetic and epigenetic alterations. These changes involve mutations in critical genes responsible for regulating essential cellular processes. The initial step involves *APC* gene mutation, which disrupts cell proliferation leading to adenoma’s development. Subsequently, *KRAS* and *BRAF* mutations play a crucial role driving the shift towards advanced adenomas by affecting cell cycle progression, particularly cell growth and proliferation. Finally, *TP53* loss of function occurs during the transition from advanced adenoma to adenocarcinoma. This progressive sequence of genetic alterations contributes to the transformation of normal colon cells into cancerous cells. Image created using BioRender.com (accessed on 7 May 2023).

**Figure 2 ijms-24-11023-f002:**
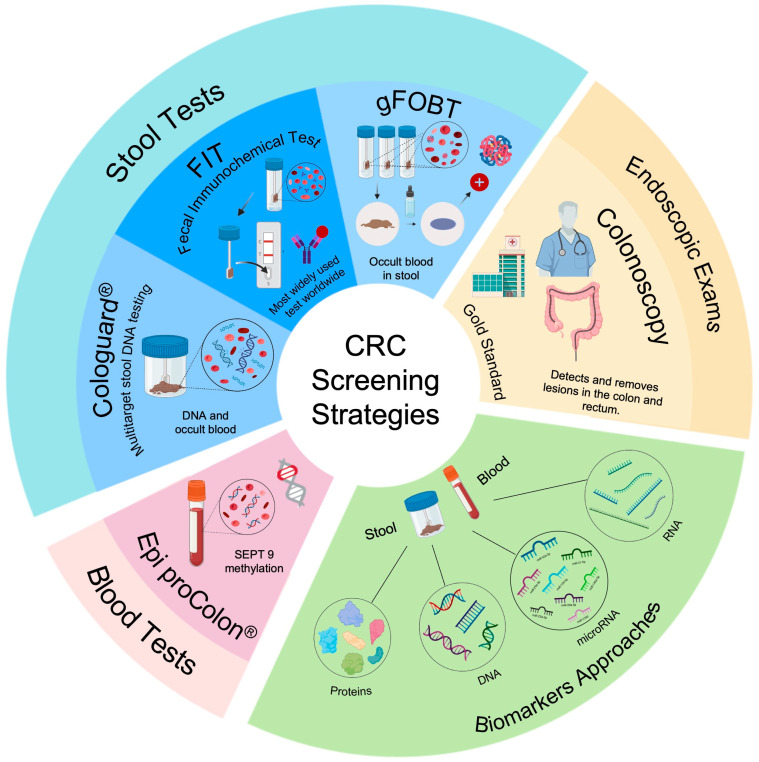
CRC screening strategies. Current approaches for CRC diagnosis encompass stool tests, such as FIT, gFOBT, and Cologuard^®^, as well as blood tests, such as Epi proColon^®^, while colonoscopy remains the gold-standard examination. In recent years, a biomarker-based approach has emerged, utilizing molecules, such as microRNAs, RNA, proteins, and DNA, with the potential to enhance the development of more effective CRC screening methods. Image created using BioRender.com (accessed on 30 January 2023).

**Figure 3 ijms-24-11023-f003:**
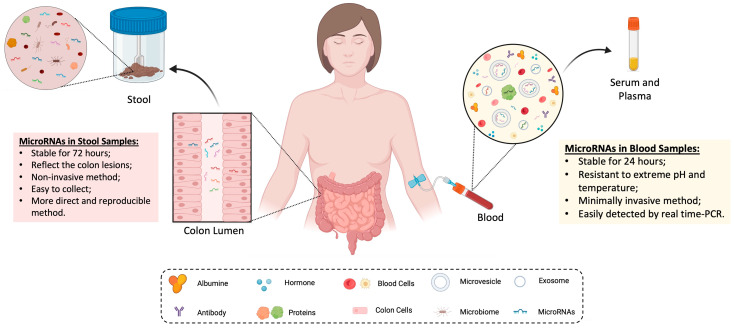
Interplay of MicroRNAs in Blood and Stool samples. MicroRNAs serve as vital carriers of genetic information between cells within biological samples. In blood samples, microRNAs are expressed in serum and plasma, as they are actively secreted by colon cells into the circulatory system. In stool samples, microRNAs are not only secreted into the lumen, but also present due to the continuous shedding and exfoliation of colon cells. Image created using Biorender.com (accessed on 7 May 2023).

**Figure 4 ijms-24-11023-f004:**
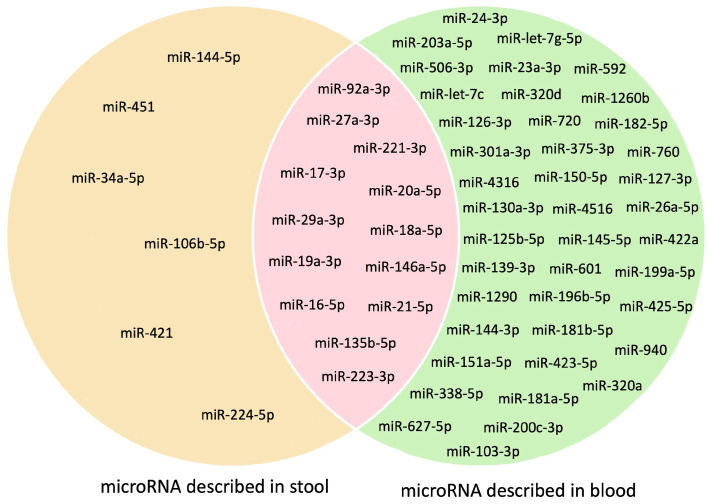
MicroRNAs as Potential Biomarkers for CRC Non-Invasive Screening. This figure illustrates the distribution of microRNAs considered useful for CRC non-invasive screening in stool (left side) and blood (right side), as well as those present in both sample types (center). Out of the total of 104 microRNAs studied, 60 have shown potential as biomarkers for CRC. The majority of these microRNAs (68%) have been described in blood samples, indicating a growing focus on this sample type. In contrast, only 10% of these potential microRNAs were explored specifically in stool samples. Moreover, 22% of the microRNAs have been analyzed in both blood and stool samples, highlighting their potential utility across different specimen types. Abbreviation: miR—microRNA.

**Figure 5 ijms-24-11023-f005:**
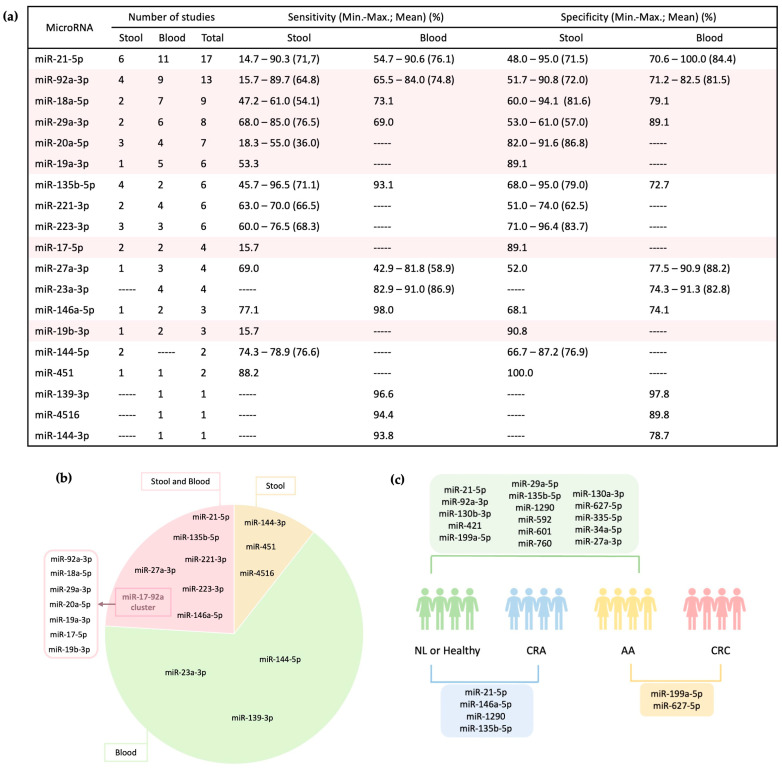
Promising microRNAs with clinical relevance in CRC diagnosis. (**a**) This panel displays the performance of microRNAs in CRC detection, along with the corresponding number of articles in which they are described in the literature. Notably, microRNAs belonging to the miR-17-92a cluster are highlighted in color. (**b**) This section illustrates the distribution of evaluated sample types in which the microRNAs with best performance results were analyzed. (**c**) This segment highlights the comparison study groups used to assess the performance of the microRNAs described detecting CRC and precancerous lesions. Abbreviations used: miR—microRNA; NL—No Lesion; CRA—Colorectal Adenoma; AA—Advanced Adenoma; CRC—Colorectal Cancer.

**Figure 6 ijms-24-11023-f006:**
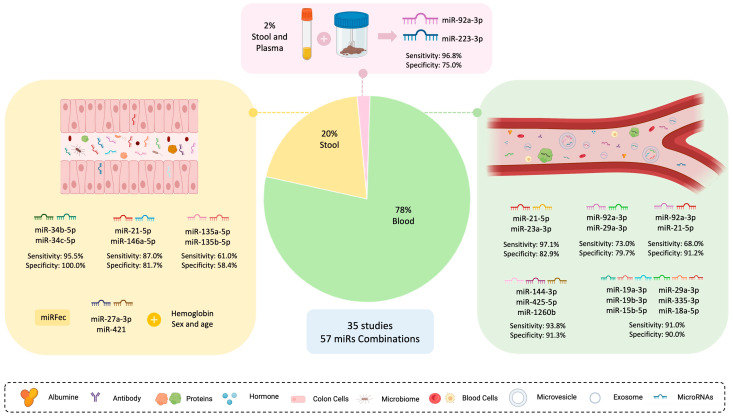
Promising Combinations of MicroRNAs for Enhanced CRC Detection. These combinations comprise microRNAs with distinct expression patterns and have been extensively studied in both stool and blood samples. Notably, one combination includes two microRNAs with divergent expression levels in stool and plasma specimens, which are analyzed together. The majority of combinations were evaluated in blood samples and exhibit high sensitivities and specificities. In contrast, 20% of microRNA combinations are described in stool samples, including the miRFec algorithm, enabling the detection of precancerous lesions, such as advanced adenoma. Abbreviation: miR—microRNA. Image created using Biorender.com (accessed on 31 May 2023).

## Data Availability

Not applicable.

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
