# Peer review of "MicroRNA Biomarkers as Promising Tools for Early Colorectal Cancer Screening—A Comprehensive Review"

_ijms, 2023, doi:10.3390/ijms241311023_

Round 1

Reviewer 1 Report

Thank you for the opportunity to review this paper. 

Overall, it is an extensive review of the current state of research on the role of microRNAs as a non-invasive method for the diagnosis of colorectal cancer. It is a very interesting topic because colorectal cancer (CRC) is one of the most prevalent cancers and a leading cause of cancer death worldwide. Less than half of cases are diagnosed when the cancer is locally advanced, so the identification of diagnostic biomarkers remains a challenge.

The paper is well structured and written, and has up to date references. 

I have some suggestions:

Regarding the review of the existing literature you must specify the inclusion and exclusion criteria. Also specify all databases, registers and other sources searched or consulted to identify studies and it is advisable to specify the date when each source was last searched or consulted.

At the final section:

Strengths and limitations of the review should be clearly discussed. Discuss any limitations of the review processes used and implications of the results for practice and future research. It would also be interesting to compare with alternative biomarkers that are also promising as diagnostic tools in colorectal cancer.

None

Author Response

The authors would like to thank the reviewers’ comments for providing your feedback on the paper.

We appreciate all the suggestions, and we would like to address them accordingly.

Comment 1: “Regarding the review of the existing literature, you must specify the inclusion and exclusion criteria. Also specify all databases, registers and other sources searched or consulted to identify studies and it is advisable to specify the date when each source was last searched or consulted.”

Response 1: Regarding the review of the existing literature, I acknowledge the importance of specifying the inclusion and exclusion criteria. To enhance the transparency and reproducibility of the study, we will ensure that the paper clearly outlines the criteria used for including or excluding relevant studies. Additionally, we will provide information about the databases, registers, and other sources searched or consulted to identify studies, and we have specified the date when each source was last searched, thus enabling readers to understand the currency of the information obtained. We added these information in chapter 2.

Comment 2: “Strengths and limitations of the review should be clearly discussed. Discuss any limitations of the review processes used and implications of the results for practice and future research. It would also be interesting to compare with alternative biomarkers that are also promising as diagnostic tools in colorectal cancer”.

Response 2: In the final section of the paper, we included a more deep discuss about the strengths and limitations of the review. We understand the significance of recognized the limitations of the review processes used. By doing so, we can provide readers with a comprehensive understanding of the potential biases or constraints associated with the study. Moreover, we had emphasized the implications of the results for practice and future research, highlighting the practical applications and areas for further investigation.

We appreciate your suggestion to compare with alternative biomarker, and had included the aspect in chapter 5, allowing for a more comprehensive analysis of the topic.  By including such a comparison, we can provide readers with a broader perspective on the potential diagnostic options available.

Reviewer 2 Report

First chapter of the review relates to the tests currently avaialble for early-stage detection of colorectal cancer. This is a much-needed introduction and also highlights the current limitations of each one of these tests or procedures. 

The 2nd part of chapter 1 introduces the nature of miRNAs - a very general but necessary short introduction. Again, their pros and cons for screening and CRC in particular are outlined. 

Chapter 2 focuses initially on WHERE to find miRNAs for screening and diagnostics (which to me, surprisingly, includes stool samples), and I think nothing has been left out. 

Generally, the authors have done an amazingly solid analysis of the existing literature for each one of the topics discussed in more detail in the review. It leaves the impression of a really thorough literature research. But looking at the reference lit, with "only" 123 papers cited, I think this also shows considerable focus in selecting that literature. With less stringent focus, this could have been easily expanded to 200-300 publications (623 publications in 2022 alone in Pubmed on miRNAs in colorectal cancer). 

What I also find convincing, or even impressive, is the thorough analysis of WHICH miRNAs have been identified in which studies and in which samples. This mata-analysis of multiple studies is potential very valuable to settle on "hot spots" of informative miRNAs for diagnostic purposes, possibly at the same time also with interesting functions in CRC progression. I like the effort the authors make to NOT only look at the purposes of diagnostics but also look into the biology of both the disease and the markers. 

The description of individual miRNAs of outstanding interest in chapter 3 often gets a bit difficult to read... there are more numbers than letters in some of these sections. While the previous chapters have been or rather general interest, the upcoming sections are more of specialist interest - those who work with them. These chapters are a bit "chewy" to read through. On the other hand, these sections also demonstrate the level of detail these authors have tried to capture, and that's fully justified... 

The images do a good job in trying to summarize the "who is who" of miRNAs in CRC development and screening. nevertheless, the text is still a biot of a challenge. I wonder if some of this very detailed info could be captured and summarized in tables? There arent too many tables in this manuscript. 

In summary, I dont have much advise to the authors how to further improve the review. I think its quite mature, truly comprehensive, sometimes it goes much into details - but always and overall informative. My respect to the authors. 

Grammar is mostly okay, there are a bunch of little mistakes, they could also be typos, but I dont see a problem getting this fixed. 

Author Response

We would like to express our gratitude for taking the time to review the first chapter of the review and for sharing your thoughts and observations. We appreciate your positive feedback and would like to address some points you have raised.

Comment 1: “Generally, the authors have done an amazingly solid analysis of the existing literature for each one of the topics discussed in more detail in the review. It leaves the impression of a thorough literature research. But looking at the reference lit, with "only" 123 papers cited, I think this also shows considerable focus in selecting that literature. With less stringent focus, this could have been easily expanded to 200-300 publications (623 publications in 2022 alone in Pubmed on miRNAs in colorectal cancer).” 

Response 1: Regarding the number of references cited, we understand your observation about the focused selection of literature. We aimed to include the most relevant and influential publications to provide readers with a solid foundation for understanding the topic.

Comment 2 “What I also find convincing, or even impressive, is the thorough analysis of WHICH miRNAs have been identified in which studies and in which samples. This mata-analysis of multiple studies is potential very valuable to settle on "hot spots" of informative miRNAs for diagnostic purposes, possibly at the same time also with interesting functions in CRC progression. I like the effort the authors make to NOT only look at the purposes of diagnostics but also look into the biology of both the disease and the markers.” 

Response 2: Thank you for your comments.

Comment 3: “The description of individual miRNAs of outstanding interest in chapter 3 often gets a bit difficult to read... there are more numbers than letters in some of these sections. While the previous chapters have been or rather general interest, the upcoming sections are more of specialist interest - those who work with them. These chapters are a bit "chewy" to read through. On the other hand, these sections also demonstrate the level of detail these authors have tried to capture, and that's fully justified...” 

Response 3: We appreciate your feedback regarding the readability of the description of individual miRNAs in section 3. We understand that these sections may be more specialized and denser with technical information. However, we believe that capturing such detailed information is crucial for researchers working in the field. Nevertheless, we highlighted this as a limitation of the review in the Final considerations section.

Comment 4: “The images do a good job in trying to summarize the "who is who" of miRNAs in CRC development and screening. nevertheless, the text is still a bit of a challenge. I wonder if some of this very detailed info could be captured and summarized in tables? There aren’t too many tables in this manuscript.” 

Response 4: Your suggestion to utilize additional tables to summarize the detailed information is well-received. We had explored the incorporation of tables to enhance the accessibility and organization of the content, providing readers with a clearer overview of the key information presented. However, since the paper is a comprehensive and exhaustive review of the literature, we opted to use figures to make it more visually appealing to readers. Nonetheless, we have submitted two comprehensive tables as supplementary data, highlighting and summarizing all the important data collected from the literature in a more systematic manner. This approach gives specialist readers the opportunity to access and use the information effectively. 

  Comments on the Quality of English Language

"Grammar is mostly okay, there are a bunch of little mistakes, they could also be typos, but I dont see a problem getting this fixed. "

We corrected some typos and little mistakes that we found.